# Characterization and Pathogenicity of *Mannheimia glucosida* Isolated from Sheep

**DOI:** 10.3390/microorganisms13122676

**Published:** 2025-11-25

**Authors:** Qibing Gu, Min Gao, Taichun Gao, Youwen Yang, Xue Sha, Falong Yang

**Affiliations:** 1College of Animal & Veterinary Sciences, Southwest Minzu University, Chengdu 610041, China; guqbing@swun.edu.cn (Q.G.); 13550059485@163.com (M.G.); gseason1353@163.com (T.G.); 2Department of Agricultural and Biosystems Engineering, South Dakota State University, Brookings, SD 57007, USA; yangyouwen.yang@jacks.sdstate.edu (Y.Y.); shaxue.sha@jacks.sdstate.edu (X.S.)

**Keywords:** respiratory diseases, *Mannheimia glucosida*, biochemical characterization, virulence, drug resistance

## Abstract

Bacteria of the genus *Mannheimia* are major pathogens of respiratory diseases in ruminants and pose a significant threat to the global ruminant industry. However, the biological characteristics and pathogenic mechanisms of *Mannheimia glucosida* remain unclear. In this study, we isolated five strains of *M. glucosida*, which specifically hydrolyzed esculin, from sheep with respiratory disease in China. All five strains of *M. glucosida* were found to encode the adhesion-related gene *adh* and the anti-phagocytosis-related gene *plpD*, as determined by a virulence gene assay. Moreover, all *M. glucosida* isolates were resistant to streptomycin. Phylogenetic analysis based on *16S rRNA*, *infB*, and *sodA* genes showed that the *sodA* gene could be a valuable indication for the analysis of bacterial genetic evolution in the genus *Mannheimia*. By mouse modeling, *M. glucosida* D251 was further found to cause multiorgan damage with an LD_50_ of 1.35 × 10^6^ CFU. Meanwhile, by combining whole genome sequencing with bioinformatic analysis, we found that the D251 genome encodes a large number of virulence and drug resistance genes. Finally, we established a highly sensitive and specific PCR assay for *M. glucosida*. Collectively, these results indicate that *M. glucosida* may be an important pathogen in respiratory disease in sheep in China and provides a theoretical basis for the clinical diagnosis and treatment of this disease.

## 1. Introduction

Sheep are an economically and culturally important farm animal. Respiratory diseases are the most common and serious threats to large-scale sheep farms worldwide [1]. Currently, various pathogens are known to be responsible for respiratory diseases in sheep. These include *Mannheimia haemolytica*, *Pasteurella multocida*, parainfluenza virus type 3, respiratory syncytial virus, and *Mycoplasma ovipneumoniae* [2]. An increasing number of studies indicate that bacteria belonging to the genus *Mannheimia* are significantly implicated in respiratory diseases affecting sheep [3,4,5]. The genus *Mannheimia* comprises five distinct species: *M. haemolytica*, *Mannheimia glucosida*, *Mannheimia ruminalis*, *Mannheimia granulomatis*, and *Mannheimia varigena* [6]. Among these pathogens, *M. haemolytica* has been the subject of extensive research, as it is recognized not only as the primary causative agent of “shipping fever” in cattle but also as a significant pathogen responsible for pneumonia in sheep and adult cattle [7,8]. Nevertheless, research on *M. glucosida* has been comparatively underdeveloped, and its biological characteristics and pathogenic potential remain inadequately elucidated.

*M. glucosida* was initially classified as the A11 serotype of *M. haemolytica*. However, in 1999, Angen et al. redefined it as *M. glucosida* based on findings from DNA-DNA hybridization and 16S rRNA sequencing analyses [6]. The primary distinction between this pathogen and other members of the *Mannheimia* genus lies in its ability to produce β-glucosidase, an enzyme capable of hydrolyzing esculin [9]. *M. glucosida* was initially isolated from the nasal cavities of healthy cattle and sheep [10]. However, in 2002, Angen et al. isolated a strain of *M. glucosida* from a sheep with pneumonia [11]. Furthermore, Omaleki et al. isolated nine strains of *M. glucosida* from sheep with mastitis [12]. The findings of these studies indicate that *M. glucosida* may represent a potential pathogen warranting further exploration of its pathogenicity.

The use of antibiotics is the mainstay of treatment for respiratory disease caused by bacteria of the genus *Mannheimia*, which are common opportunistic pathogens of the respiratory tract. Research has been conducted on the identification of drug-resistance genes in *M. haemolytica* isolates obtained from cattle suffering from respiratory disease [13]. The findings show that these isolates possess multiple resistance genes, which contribute to an elevated level of antimicrobial resistance [13]. Similarly, 73 strains of *M. haemolytica* isolated from healthy sheep were resistant to penicillin and 9 strains were resistant to chlortetracycline and oxytetracycline [14]. The rise in drug resistance among bacteria belonging to the genus *Mannheimia* has been linked to the presence of mobile genetic elements, which have demonstrated the capacity for transfer between *Pasteurella multocida*, *M. haemolytica*, and *Escherichia coli* [15]. However, drug resistance in *M. glucosida*, an important bacterium in the genus *Mannheimia*, has not been studied, and this will be one of the focuses of future work.

The detection of bacteria of the genus *Mannheimia* is particularly important for the treatment of respiratory diseases. Various assays have been developed, such as polymerase chain reaction (PCR), multiplex PCR, fluorescent quantitative PCR, and recombinase polymerase amplification (RPA), each designed to specifically identify one or multiple pathogenic bacterial species. Kumar et al. established a triple PCR assay targeting specific genes of *M. haemolytica*, which enables the specific identification of *M. haemolytica* isolates [16]. Furthermore, a fluorescent quantitative PCR technique has been developed that enables the specific identification of bacteria belonging to the genus *Mannheimia* [17]. Four pathogens, *M. haemolytica*, *P. multocida*, *Histophilus somni*, and *Mycoplasma bovis*, can be detected using the RPA method [18]. Although multiple assays have been established, methods for the specific detection of *M. glucosida* are still needed.

In this study, five strains of *M. glucosida*, characterized by their specific ability to hydrolyze esculin, were isolated from sheep exhibiting respiratory disease. Subsequently, comprehensive biochemical characterization, virulence assessment, and drug resistance analysis of *M. glucosida* were conducted. The pathogenicity of isolate D251 was validated through a mouse model, which demonstrated its capacity to induce multi-organ damage. These results suggest that *M. glucosida* may be an important pathogen in respiratory disease in sheep. Meanwhile, we sequenced the whole genome of D251 and established a species-specific PCR assay for *M. glucosida*. This will provide novel ideas for the rapid diagnosis of clinical respiratory diseases.

## 2. Materials and Methods

### 2.1. Collection of Clinical Samples

Nasal cotton swabs were obtained from thirty adult sheep exhibiting pronounced respiratory symptoms, including cough, nasal discharge, dyspnea, and fever. The entire sheep farm has a total of 2000 sheep. These samples were subsequently inoculated onto whole blood agar plates for further analysis.

### 2.2. Isolation and Cultivation of Strains

Thirty whole blood agar plates were placed in a 37 °C incubator (normal atmospheric environment) for 20 h, and then a single colony of suspected *Mannheimia* bacteria was streaked onto TSA containing 5% horse serum for purification. After three rounds of purification, single purified colonies were picked for Gram staining microscopy. The confirmed single colony was inoculated into 2 mL of TSB containing 5% horse serum for bacterial enrichment and cultured at 37 °C with constant shaking for 8 h. The purified enrichment was used for DNA extraction and PCR detection.

### 2.3. Extraction of Total DNA

Total DNA was extracted from the isolated strains utilizing the phenol-chloroform extraction method. The resulting DNA samples were subsequently stored at a temperature of −20 °C.

### 2.4. PCR Identification of Mannheimia

In order to determine whether the isolated strain is a member of *Mannheimia*, we reference the multiple PCR method of *Mannheimia* established by Alexander [19]. This method can detect *M. haemolytica*, *M. glucosida*, and *M. ruminalis*. The primer sequences are shown in Appendix A. The reaction program is 95 °C for 5 min; 35 cycles of 94 °C for 30 s and annealing temperature for 30 s, and 72 °C for 40 s; 72 °C for 10 min, 4 °C to end the reaction. The PCR products were analyzed by 1.5% agarose gel electrophoresis, and the results were observed in the gel imaging analysis system.

### 2.5. Biochemical Test of Isolated Strains

According to the characteristics of *M. glucosida* [9], which can hydrolyze esculin, this study will carry out the hydrolysis of esculin to determine how many of the strains identified as *Mannheimia* by multiple PCR are *M. glucosida*. To further analyze the biochemical characteristics of the isolated *M. glucosida*, we placed the isolated strains in the VITEK 2 automatic biochemical identification instrument for additional biochemical research.

### 2.6. Virulence Gene Detection

To detect the important virulence genes carried by *M. glucosida*, the virulence gene PCR assays established by Klima et al. and García-Alvarez et al. were referenced [20,21]. The PCR assays were carried out for the virulence genes *gcp*, *gs60*, *tbpA*, *tbpB*, *lktC*, *nmaA*, *adh*, and *plpD*. The primer information is shown in Appendix A.

### 2.7. Antimicrobial Susceptibility Testing

The antimicrobial susceptibility of the *M. glucosida* isolates was assessed using the Kirby-Bauer disk diffusion method according to CLSI guidelines [22]. Firstly, the purified single colony was inoculated into TSB medium containing 5% horse serum and incubated at 37 °C for 8 h. Secondly, the bacterial concentration was adjusted to 1.5 × 10^8^ colony-forming units (CFU)/mL using TSB enrichment solution. Then, 100 μL of the diluted bacterial solution was pipetted and uniformly spread onto TSA plates containing 5% horse serum using a sterilized applicator stick. The sensitized tablets were placed with a 2 cm interval between each plate for the sensitization test. The disc concentrations for the eight antibiotics are as follows: Florfenicol, 30 μg; Cephalothin, 30 μg; Doxycycline, 30 μg; Cephalexin, 30 μg; Streptomycin, 10 μg; Kanamycin, 30 μg; Gentamicin, 30 μg; and Cefoxitin, 30 μg. The plates were inverted and incubated at 37 °C for 20 h, and the results were assessed based on the inhibitory effect of the bacteria. The drug sensitization results were classified into three categories: sensitive (S), intermediary (I), and resistant (R), according to the diameter of the inhibition zone.

### 2.8. Analysis of the Pathogenicity of M. glucosida in Mice

To ascertain the 50% lethal dose (LD_50_), we employed six concentrations ranging from 1.0 × 10^4^ CFU/mL to 1.0 × 10^9^ CFU/mL and conducted experiments on a cohort of 30 specific pathogen-free (SPF) BALB/c mice, aged between 6 and 8 weeks. Mice were used instead of sheep to conduct preliminary pathogenicity screening under controlled conditions. Before the test, the mice were kept under normal conditions for 3 days, during which their activities and mental status were observed and normalized. Each concentration was injected intraperitoneally into 5 mice at a volume of 0.5 mL per mouse. Simultaneously, 5 mice in the control group were injected with PBS using the same inoculation method and dose. The morbidity and mortality of mice in each group were observed and recorded at all times, and the LD_50_ was calculated using the modified Koch’s method. Concurrently, the deceased mice were subjected to dissection, during which the liver, spleen, lungs, and kidneys were systematically collected. These organs were subsequently fixed in 4% paraformaldehyde and forwarded to Chengdu Rilai Biotechnology Co. for the preparation of tissue sections and hematoxylin and eosin (HE) staining, facilitating the examination of pathological alterations. In addition, to analyze the infestation of each organ in the mice by the strain, we extracted total DNA from each organ of the deceased mice and tested it against *M. glucosida*.

### 2.9. Sequence Analysis of the lktA Gene

To investigate the molecular characterization of the *lktA* gene from the isolated strain of *M. glucosida*, three pairs of primers (Appendix A) that were previously developed in our laboratory were employed for the segmental amplification of the complete *lktA* gene. The resulting PCR products were subsequently dispatched to Sangon Bioengineering (Shanghai) Co. (Shanghai, China) for further analysis. The SeqMan software (https://www.dnastar.com/, accessed on 16 April 2023) was employed to sequence and splice the fragments of the *lktA* gene in order to obtain the complete coding sequence (CDS). Additionally, a homology analysis of the *lktA* gene across five isolates was conducted utilizing the BLAST tool (https://blast.ncbi.nlm.nih.gov/Blast.cgi, accessed on 18 April 2023) provided by the National Center for Biotechnology Information (NCBI). Furthermore, an evolutionary tree was constructed using the MAGA software, version 7.0.26.

### 2.10. Whole Genome Sequencing, Splicing and Assembly

The whole genome sequencing of *M. glucosida* was completed by Shanghai Lingen Biotechnology Co., Ltd. (Shanghai, China).

### 2.11. Establishment of Species-Specific PCR for M. glucosida

Since *M. glucosida* can hydrolyze esculin, and the enzyme involved in hydrolyzing esculin is β-glucosidase, this study refers to the gene encoding β-glucosidase (*bglA*) of *M. glucosida* D251 strain. The fragment size is 758bp, sequence is following: bglA-F: 5′-ATGAAATTCCGTTGGGCTTAG-3′; bglA-R:5′-CTTTATCGTAAGCACCCAGTCC-3′. To determine the optimal annealing temperature, the following temperatures were tested: 51 °C, 53 °C, 55 °C, 57 °C, 59 °C, and 60 °C.

### 2.12. Sensitivity and Specificity Analysis of Species-Specific PCR

The optimized PCR conditions were used to amplify the genomic DNA of *M. haemolytica*, *M. ruminalis*, *P. multocida*, *M. ovispneumoniae*, and *E. coli* as templates for the PCR specificity test. The amplified products were subjected to 1.5% agarose gel electrophoresis, and the results were observed using a gel imaging analysis system.

To study the lower limit of detection of the bacterial genome by this method, the concentration of the genomic DNA of the *M. glucosida* D251 strain was measured with a nucleic acid protein detector. According to its genome size, an online tool (https://cels.uri.edu/gsc/cndna.html, accessed on 12 December 2023) was used to calculate the copy number. Subsequently, a 10-fold gradient dilution was performed to prepare different concentrations of DNA.

Furthermore, to study the lower limit of detection of colony-forming units (CFU) in the bacterial solution using this method, we employed the plate count method to determine the concentration of the isolated strain D251 and performed a 10-fold gradient dilution. The phenol-chloroform method was utilized to extract DNA from each bacterial solution, and the DNA template was then amplified under optimized PCR conditions. The amplified products were analyzed by 1.5% agarose gel electrophoresis, and the results were observed in the gel imaging analysis system.

### 2.13. Identification of Clinical Samples by Species-Specific PCR

In order to evaluate the detection rate of the established PCR method for *M. glucosida* in clinical samples and to initially understand the prevalence of *M. glucosida* in sheep, we analyzed samples from 80 individuals in Gansu Province, Qinghai Province, and Ganzi Prefecture, Sichuan, China. Samples of sheep nasal swabs were extracted using a phenol-chloroform method for DNA extraction and tested according to the established PCR method.

## 3. Results

### 3.1. Isolation and Identification of M. glucosida

To better understand the prevalence of *M. glucosida* in sheep, bacterial isolation was first conducted. Single colonies were obtained by inoculation on blood agar plates and purified three times. The identification of *M. glucosida* was conducted using PCR techniques targeting the genus *Mannheimia*, in conjunction with the esculin hydrolysis assay. Five strains (D251, G2, G3, G4, G5) of *M. glucosida* were successfully obtained through isolation procedures. Isolated *M. glucosida* exhibits the formation of smooth and glossy colonies when cultured on blood agar plates (Figure 1A). The primer sets LKT, LKT2, and HP all specifically amplified *M. glucosida* DNA fragments with the correct sizes (Figure 1B–D). Furthermore, it was observed that *M. glucosida* possesses the unique ability to hydrolyze esculin, a characteristic that is not exhibited by either *M. haemolytica* or *M. ruminalis* (Figure 1E). The findings suggest that all five isolates belong to the *M. glucosida* species.

### 3.2. Biochemical Characteristics of M. glucosida

The analysis of biochemical properties revealed that all five isolates tested positive for β-glucosidase and esculin (Table 1). In contrast, both *M. haemolytica* and *M. ruminalis* showed negative results. Furthermore, all isolates exhibited positive reactions to the ADO (Adonitol), dCEL (D-cellobiose), d-GLU (D-glucose), d-MAN (D-mannitol), and SAC (Sucrose) (Appendix A). These results will provide a strong basis for the identification of *M. glucosida*.

### 3.3. Pathogenicity Assays of M. glucosida in Mice

Through virulence-associated gene testing, the *adh*, *plpD*, *tbpA*, *tbpB*, *gcp*, *gs60*, and *lktC* genes were specifically amplified in the isolates of *M. glucosida*, suggesting a possible strong virulence of the isolates (Figure 2A). In the five strains of *M. glucosida*, the detection rate was 100% for the adhesion-related *adh* and colonization-related *plpd* genes, 80% for *tbpA*, *tbpB*, *gcp*, *gs60*, and *lktC*, and 0% for the *nmaA* gene (Table 2). These results suggest that *M. glucosida* may have a strong ability to adhere and colonize in vivo. To further understand the pathogenicity of *M. glucosida*, strain D251 was selected for analysis. The LD_50_ of D251 was determined to be 1.35 × 10^6^ CFU/mL through experimentation utilizing a murine model of infection (Table 3). Pathological examinations indicated the presence of multiple organ injuries in the mice that succumbed to D251 (Figure 2B). In the dead mice, there was hepatocellular steatosis in the liver, splenic sinus stasis in the spleen, inflammatory cell infiltration with predominantly rod-nucleated neutrophils in the lungs, and glomerular necrosis in the kidneys. Furthermore, *M. glucosida* was detected in different organs of mice (Figure 2C). These results suggest that *M. glucosida* has a broad colonization ability and strong pathogenicity in mice, but its pathogenic potential in ruminants needs to be further investigated.

### 3.4. Antimicrobial Resistance Analysis of M. glucosida

To evaluate the susceptibility of the isolates to clinically used antibiotics, a disc susceptibility test was performed. The results revealed that all five strains of *M. glucosida* exhibited resistance to streptomycin while demonstrating susceptibility to florfenicol, ceftiofur, and cefoxitin (Table 4). In addition, *M. glucosida* showed intermediate resistance to aminoglycoside antibiotics. These data may provide some theoretical basis for the treatment of clinical infections caused by *M. glucosida*.

### 3.5. Phylogenetic Analysis of M. glucosida

Further phylogenetic analysis of *lktA* genes and housekeeping genes was conducted to characterize the genetic evolution of *M. glucosida* [23]. The *lktA* genotype analysis was first performed by retrieving *lktA* gene information from GenBank (Appendix A). It was found that five isolates belonged to the *lktA*4 genotype (Figure 3A). The G3 strain belonged to the *lktA*4.5 subtype, while the G4 and G5 strains belonged to the *lktA*4.3 subtype. D251 (indicated with G1 in Figure 3) and G2 were not associated with any branch and were classified as belonging to novel subtypes designated *lktA*4.7 and *lktA*4.8, respectively. Subsequently, information on housekeeping genes was searched for in the GenBank database (Appendix A), and a phylogenetic analysis was performed. Based on the *16S rRNA* and *infB* gene, *M. glucosida* is in the same large branch as *M. haemolytica*, suggesting that it is more closely related to *M. haemolytica* (Figure 3B,C). It is also suggested that the *16S rRNA* gene and the *infB* gene exhibit a higher degree of conservation in *M. glucosida* and *M. haemolytica*, rendering them unsuitable for the identification of bacterial species within the genus *Mannheimia*. Based on the *sodA* gene, all *M. glucosida* are in the same branch and are distantly related to the other bacteria of *Mannheimia* spp (Figure 3D). The results show that the *sodA* gene varies greatly between different *Mannheimia* species. *SodA* could be a better target for developing diagnostic assays compared to the more conserved *16S rRNA* and *infB* genes.

### 3.6. Whole-Genome Sequencing Analysis of M. glucosida

Due to the current lack of genomic information on *M. glucosida*, strain D251 was further selected for whole genome sequencing. The NCBI accession number for the D251 genome is CP176502. The D251 genome size is 2.4 Mb with 41% GC content (Figure 4A). There are 2279 genes on the D215 genome, of which 2183 are protein-coding genes. The COG functional classification of genomic proteins revealed that the genome of D251 encodes a large number of proteins with unknown functions in addition to metabolism-related proteins. Exploring the biological functions of these unknown proteins will help us understand the pathogenic properties of *M. glucosida* (Figure 4B). The genes carried on the D251 genome were categorized into five classes by KEGG annotation: (A) Metabolism, (B) Genetic Information Processing, (C) Environmental Information Processing, (D) Cellular Processes, and (E) Organismal Systems (Figure 4C). 79.5% (2065/2597) of these genes are involved in metabolic processes.

### 3.7. Prediction of Virulence and Resistance Genes in M. glucosida

Prediction of virulence genes based on the D251 whole genome sequence using the VFDB online database (http://www.mgc.ac.cn/VFs/, accessed on 3 February 2024). A total of 96 virulence genes were predicted in D251, of which 29 were endotoxin-associated virulence genes, 23 were iron uptake-associated virulence genes, and 14 and 11 were associated with immunomodulation and adhesion, respectively (Appendix A). These data suggest that the pathogenesis of *M. glucosida* D251 may involve a complex process, warranting additional comprehensive research. CARD (https://card.mcmaster.ca/, accessed on 6 May 2024) prediction revealed that multiple resistance genes were encoded in the D251 genome, including five fosfomycin resistance genes, three fluoroquinolone resistance genes, two neomycin resistance genes, and one streptomycin resistance gene, respectively (Table 5). The results suggest that *M. glucosida* D251 may be resistant to a variety of antibiotics, and further determination of its resistance phenotype by drug sensitivity testing with more antibiotics is needed. Furthermore, in combination with the previous analysis of the drug sensitivity test, it is suggested that the predicted streptomycin resistance gene may mediate the resistance of D251 to streptomycin.

### 3.8. Establishment of the Specific Detection for M. glucosida

In response to the absence of an efficient detection technique for *M. glucosida*, we developed a species-specific PCR method. Since β-glucosidase, the product of the *bglA* gene encoded in *M. glucosida*, is involved in the hydrolysis of esculin, specific amplification primers for *bglA* were designed. Amplification of the corresponding genes from five isolates using this primer resulted in a single band of the expected size (Figure 5A). The results show that the primer can be used for the detection of *M. glucosida*. The PCR protocol was optimized through the application of a gradient annealing temperature, resulting in the identification of 59 °C as the optimal annealing temperature (Figure 5B). To verify the specificity of the optimized PCR method, various bacterial strains were tested. The results indicated that amplification was observed exclusively for *M. glucosida*, whereas neither *M. haemolytica* nor *M. ruminalis*, among other species, exhibited amplification (Figure 5C). Furthermore, sensitivity analysis revealed that the lower limit of genomic detection for *M. glucosida* was 27.8 copies per reaction (Figure 5D). This PCR method has a lower detection limit of 56 CFU/mL for *M. glucosida* pure culture (Figure 5E). Subsequently, 80 sheep nasal swab samples from different provinces were tested using this PCR method, and the detection rate of *M. glucosida* was found to be 22.5% (18/80). The results indicate that this PCR method can be used to detect *M. glucosida* in clinical samples and also suggest that this pathogen is more widely infected in sheep (Figure 5F).

## 4. Discussion

Bacteria of the genus *Mannheimia* are important pathogens in cattle and sheep, posing a serious constraint to the development of ruminant farming [3,12]. Although *M. haemolytica* has been extensively studied, there is a dearth of research on *M. glucosida*. In this study, we have biologically characterized sheep-derived *M. glucosida*. For the first time, we have supplemented the genomic information of *M. glucosida* and established a species-specific PCR method for it.

Bacteria of the genus *Mannheimia* include *M. haemolytica*, *M. glucosida*, *M. ruminalis*, *M. granulomatis*, and *M. varigena* [6]. *M. haemolytica*, a major pathogen responsible for respiratory disease syndromes in ruminants, causes fibrotic and necrotizing lobar pneumonia, as well as pleuropneumonia [7]. The important virulence factor of *M. haemolytica* is leukotoxin (LKT), which specifically interacts with the β2 integrin receptor on leukocytes [24]. LKT promotes the release of pro-inflammatory cytokines at low concentrations [25], while at high concentrations, it causes swelling and necrosis of leukocytes, resulting in lung injury [26,27]. The *lktA* genotype of *M. haemolytica* and that of *M. glucosida* occupy distinct evolutionary branches. In the present study, the sheep-derived *M. glucosida* encodes the *lktA* gene, which produces LKTs and belongs to the *lktA*4 genotype. However, the mechanism of action of *M. glucosida* LKT in its infection is not known. In addition to LKTs, several virulence factors have been identified in *M. haemolytica*, including LPS, outer membrane proteins, and capsules [28]. In this study, all five isolates of *M. glucosida* were found to encoded virulence-associated genes *adh* and *plpD*, suggesting that *M. glucosida* exhibits a pronounced capacity for adhesion and colonization. Furthermore, the high virulence of *M. glucosida* D251 was confirmed by a mouse infection model, which resulted in multiple organ damage in mice. The results of this study suggest that *M. glucosida* may be a potentially important pathogen. Due to the limitations of cross-species pathogenicity models, future research should further utilize sheep models to investigate the pathogenicity of *M. glucosida* in sheep. If confirmed as a pathogen, *M. glucosida* likely acts as an opportunistic agent, causing disease when the host immunity is decreased or during periods of stress, such as during transport or malnutrition [29]. In contrast to *M. haemolytica*, the isolation rate of *M. glucosida* is observed to be 20–30% in healthy sheep, whereas it constitutes less than 5% in diseased animals [29]. This distribution implies that *M. glucosida* may be more closely linked to a symbiotic association with the host rather than exhibiting pathogenic tendencies. Future studies will employ metagenomic analysis to determine its specific abundance within the sheep respiratory tract microbiota and its interactions with other microorganisms, while further validating its symbiotic status through 16S rRNA sequencing.

The use of antibiotics is the primary treatment option for respiratory disease syndromes in ruminants. Antibiotic treatment often fails because bacteria carry drug-resistant genes. Multiple resistance genes have been identified in *M. haemolytica*, and some of these resistance genes are located within mobile genetic elements [30,31]. Streptomycin resistance was found in all five strains of *M. glucosida*, which may be mediated by the encoded streptomycin resistance gene. Further prediction revealed that the genome of *M. glucosida* carries multiple resistance genes, but the corresponding antibiotic resistance phenotypes need to be further validated. These results suggest that *M. glucosida* may serve as a reservoir of resistance genes that provide antibiotic resistance to respiratory pathogens. Further validation and research on the resistance of *M. glucosida* to antibiotics hold significant implications for guiding clinical antibiotic use. Furthermore, subsequent investigations will expand antimicrobial susceptibility testing to encompass additional classes of antibiotics in order to enhance the resistance profile characterization of *M. glucosida*.

Currently, for the detection of *M. glucosida*, the main methods are isolation and culture of bacteria and biochemical identification, which is time-consuming and complicated. For *M. haemolytica*, several studies have established rapid PCR detection methods [4,16,32]. In established multiplex PCR methods and fluorescent quantitative PCR methods, although *M. glucosida* is efficiently identified, bacteria such as *M. haemolytica* and *M. ruminalis* are also identified [17,19]. In this study, primers were designed to establish a PCR method using *bglA*, a gene encoding β-glucosidase on the D251 genome, based on the hydrolysis of esculin by *M. glucosida*. This PCR method specifically identifies *M. glucosida*, but not bacteria such as *M. haemolytica*, *M. ruminalis*, and *P. multocida*. The PCR method established in this study provides a theoretical basis for the rapid diagnosis of *M. glucosida*.

## 5. Conclusions

Overall, the present study was carried out to characterize the biological properties of the sheep-derived *M. glucosida* and to assess its pathogenicity in mice. In addition, the whole genome sequence analysis of *M. glucosida* was conducted for the first time, facilitating a deeper understanding of the genetic attributes of clinical isolates and their associated molecular mechanisms of resistance. A species-specific PCR method was further established to provide a rapid and reliable method for the detection of clinical *M. glucosida*. The availability of this specific diagnostic tool will facilitate faster and more accurate identification of *M. glucosida* in veterinary clinical settings, thereby aiding in timely disease management and control.

## Figures and Tables

**Figure 1 microorganisms-13-02676-f001:**
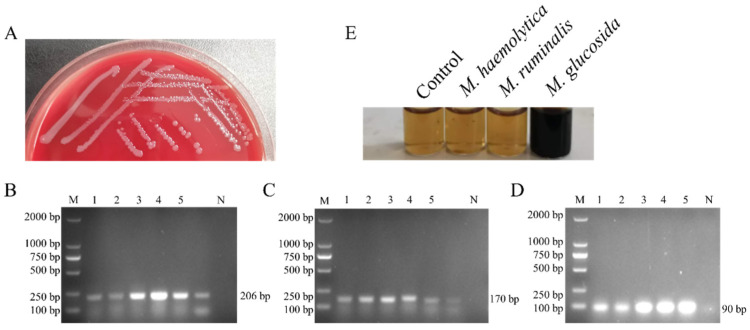
Isolation and Identification of *M. glucosida*. (**A**) Growth status of *M. glucosida* isolates on blood agar plates. Amplification results of five strains of *M. glucosida* using (**B**) LKT, (**C**) LKT2, and (**D**) HP primers. M, Marker; 1–5, D251, G2, G3, G4, G5; N, Negative control. (**E**) Esculin hydrolysis assay.

**Figure 2 microorganisms-13-02676-f002:**
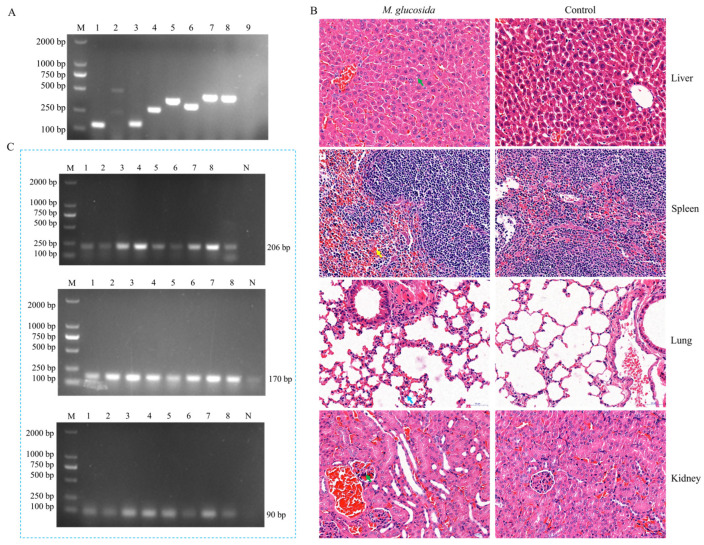
Pathogenicity assays of *M. glucosida* in mice. (**A**) Results of the virulence gene test in *M. glucosida*. (**B**) HE staining observation of each organ tissue. The green arrow in the liver indicates hepatocellular steatosis. The yellow arrow in the spleen indicates splenic sinus stasis. Inflammatory cell infiltration is indicated by the blue arrow in the lung. Glomerular necrosis is indicated by the green arrow in the kidney. Scale bar, 50 μm. (**C**) Detection of M. glucosida in different tissue organs using LKT, LKT2, and HP primers. 1: heart; 2: liver; 3: spleen; 4: lung; 5: kidney; 6: brain; 7: rectum; 8: twelve Finger intestine; N: control.

**Figure 3 microorganisms-13-02676-f003:**
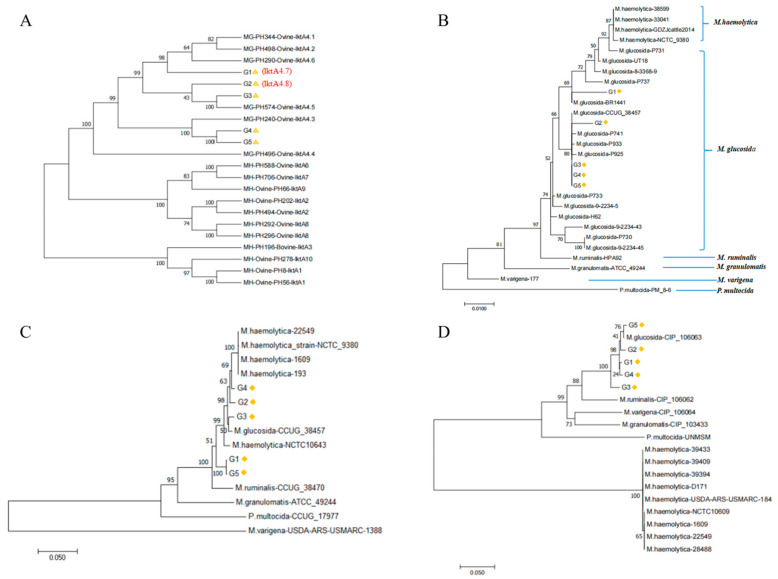
Phylogenetic analysis of (**A**) *lktA*, (**B**) *16S rRNA*, (**C**) *infB*, and (**D**) *sodA* genes in *M. glucosida*. Strain D251 is indicated by G1 in the figure.

**Figure 4 microorganisms-13-02676-f004:**
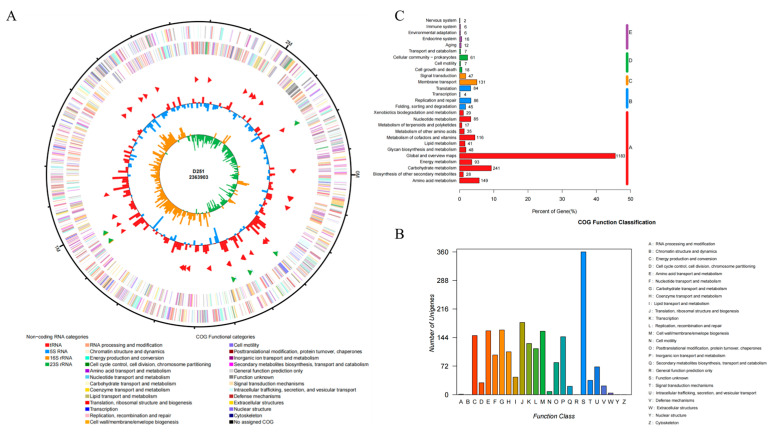
Whole genome sequencing analysis of *M. glucosida* D251. (**A**) Genome map of strain D251. The outermost circle of the map identifies the genome size; the second and third circles show the CDS on the positive and negative strands, with different colors indicating the functional classification of the different COGs of the CDSs; the fourth circle shows the rRNAs and tRNAs; and the fifth circle shows the GC content. (**B**) Statistic of genomic protein COG function. (**C**) KEGG pathway annotation. A, Metabolism; B, Genetic Information Processing; C, Environmental Information Processing; D, Cellular Processes; E, Organismal Systems.

**Figure 5 microorganisms-13-02676-f005:**
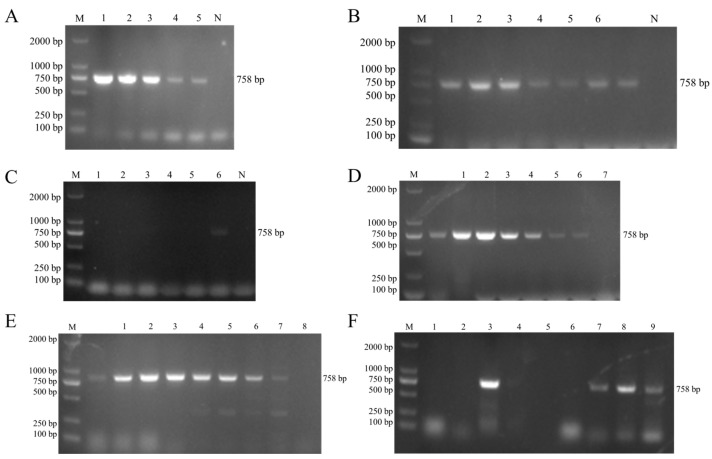
Establishment of a specific PCR detection method for *M. glucosida*. (**A**) Detection results of five isolates. 1–5: D251, G2, G3, G4, G5. N: control. (**B**) Optimization of PCR annealing temperature. 1–6: 60 °C, 59 °C, 57 °C, 55 °C, 53 °C, 51 °C. N: control. (**C**) PCR specificity analysis. 1: *E. coli*; 2: *M. haemolytica*; 3: *M. ruminalis*; 4: *P. multocida*; 5: *M. ovipneumoniae*; 6: *M. glucosida*; N: control. (**D**) Sensitivity analysis of genomic DNA for PCR detection. 1–7: 7.1 × 10^1^~7.1 × 10^−5^ ng/μL. (**E**) Sensitivity analysis of CFU for PCR detection. 1–8: 5.6 × 10^7^~5.6 × 10^0^ CFU/mL. (**F**) PCR detection of *M. glucosida* in clinical samples. 1: control. 2–9: clinical samples.

**Table 1 microorganisms-13-02676-t001:** Biochemical characterizations of different strains.

Test	Isolates					*M. glucosida*	*M. haemolytica*	*M. ruminalis*
	D251	G2	G3	G4	G5			
β-Glucosidase (NPG)	**+**	**+**	**+**	**+**	**+**	**+**	**−**	**−**
β-Xylosidase	**−**	**−**	**−**	**−**	**+**	**+/−**	**−**	**−**
L-Arabitol	**+**	**−**	**−**	**−**	**+**	**+/−**	**−**	**+**
D-Maltose	**+**	**+**	**+**	**+**	**−**	**+**	**+**	**+**
D-Sorbitol	**−**	**+**	**−**	**+**	**+**	**+/−**	**+**	**+/−**
D-trehalose	**+**	**+**	**+**	**+**	**−**	**+/−**	**UN**	**UN**
Ornithine decarboxylase	**−**	**−**	**−**	**−**	**−**	**+/−**	**−**	**−**
Esculin	**+**	**+**	**+**	**+**	**+**	**+**	**−**	**−**

**+**: Positive; **−**: Negative; **+/−**: Positive or Negative; UN: Unknown.

**Table 2 microorganisms-13-02676-t002:** Virulence profiles of *M. glucosida* strains isolated from sheep.

Virulence Genes	Isolates				
	D251	G2	G3	G4	G5
*gcp*	**+**	**+**	**+**	**+**	**−**
*gs60*	**+**	**+**	**+**	**+**	**−**
*tpbA*	**+**	**+**	**+**	**+**	**−**
*tpbB*	**+**	**+**	**+**	**+**	**−**
*lktC*	**+**	**+**	**+**	**+**	**−**
*nmaA*	**−**	**−**	**−**	**−**	**−**
*adh*	**+**	**+**	**+**	**+**	**+**
*plpD*	**+**	**+**	**+**	**+**	**+**

**+**: Positive; **−**: Negative.

**Table 3 microorganisms-13-02676-t003:** LD_50_ determination of *M. glucosida* in mice.

Dose (CFU/mL)	Deaths
1.0 × 10^4^	0
1.0 × 10^5^	0
1.0 × 10^6^	1
1.0 × 10^7^	4
1.0 × 10^8^	5
1.0 × 10^9^	5

**Table 4 microorganisms-13-02676-t004:** Antimicrobial resistance of the isolated strains.

Antimicrobial	Disk Diffusion Breakpoints (mm)		R	I	S
	R	S			
Florfenicol	≤12	≥18	0	0	100% (5/5)
Cephalothin	≤14	≥18	0	0	100% (5/5)
Doxycycline	≤12	≥16	0	20% (1/5)	80% (4/5)
Cephalexin	≤14	≥18	0	20% (1/5)	80% (4/5)
Streptomycin	≤11	≥15	100% (5/5)	0	0
Kanamycin	≤13	≥18	40% (2/5)	60% (3/5)	0
Gentamicin	≤12	≥15	20% (1/5)	80% (4/5)	0
Cefoxitin	≤14	≥18	0	0	100% (5/5)

**Table 5 microorganisms-13-02676-t005:** Genetic prediction of antibiotic resistance in *M. glucosida*.

Antimicrobial	Number of Resistance Genes	Antimicrobial	Number of Resistance Genes	Antimicrobial	Number of Resistance Genes
β-lactams	1	Vancomycin	1	Nitrofurantoin	1
Sulfonamides	1	Rifampicin	1	Isoniazid	1
Fluoroquinolones	3	Fosfomycin	5	Pyrazinamide	1
Neomycin	2	Spectinomycin	1	Daptomycin	1
Fusidic acid	1	Mupirocin	1	Streptomycin	1
Dicyclomycin	1	Coumarin	1		

## Data Availability

The original contributions presented in this study are included in the article/Appendix A. Further inquiries can be directed to the corresponding author.

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
