# Peer review of "Characterization and Pathogenicity of Mannheimia glucosida Isolated from Sheep"

_microorganisms, 2025, doi:10.3390/microorganisms13122676_

Round 1
Reviewer 1 Report
Comments and Suggestions for Authors
The paper presents the isolation, molecular characterisation, antibiotic resistance, experimental pathogenicity of Mannheimia glucosida strains from Chinese sheep, and the development of a species-specific PCR test. It does so for a poorly studied species of the genus Mannheimia, providing potentially useful genomic and diagnostic information.
The article is broadly publishable but requires substantial corrections. Although the research question is timely, the study is marred by excessive conceptual simplification, methodological weaknesses, and insufficient interpretation within the framework of Mannheimia pathobiology and sheep respiratory disease ecology. The discussion and conclusions tend to exaggerate the results disproportionately, particularly with regard to pathogenicity and epidemiological importance.
Major
The isolation of M. glucosida from diseased sheep is not entirely novel—previous isolates from pneumonia and mastitis cases have been documented (e.g., Angen et al., 2002; Omaleki et al., 2010). The novelty here lies mainly in the genomic sequencing of strain D251 and development of a PCR assay.
The claim that M. glucosida is an “important pathogen” in sheep respiratory disease is not supported by the data presented. The evidence is limited to pathogenicity in mice and detection by PCR in nasal swabs; no experimental or epidemiological evidence of causality of disease in sheep is provided.
The authors should moderate their claims and clearly differentiate between potential pathogenicity and demonstrated clinical relevance in sheep.
Only 30 nasal swabs taken from clinically affected sheep were examined, yielding five isolates. This number is too small to infer prevalence or epidemiological significance.
Nasal carriage of Mannheimia species is common in healthy animals; isolation from the respiratory tract does not imply disease causation without concurrent pathological or bacteriological evidence from lung lesions.
The Authors should discuss the ecological role of M. glucosida more critically; could it be a commensal opportunist rather than a primary pathogen?
The mouse model has limited relevance for ovine respiratory pathogens. Although it can demonstrate basic virulence potential, it cannot replicate host-specific factors such as leukotoxin–β2 integrin interactions or pulmonary immune responses.
LD₅₀ and histopathology data are presented but a quantitative comparison with M. haemolytica is lacking. Without a reference strain, it's impossible to conclude that M. glucosida is equally or more virulent.
The discussion must acknowledge the limitations of cross-species pathogenicity models and avoid over-interpretation.
The whole genome sequencing of D251 (2.4 Mb, 41% GC) is valuable, but functional annotation and comparative genomics are superficial.
The VFDB and CARD analyses are descriptive; no experimental validation or comparative analysis (e.g., with M. haemolytica or M. varigena) is provided.
Statements about “complex pathogenic processes” or “reservoirs of resistance genes” are speculative without evidence of gene expression or mobility.
Strengthen this section by:
Providing comparative genomics with closely related Mannheimia species.
Discussing which virulence genes are homologous or divergent from known factors in M. haemolytica (e.g., lktA, plpD, adh).
Avoiding claims of phenotypic resistance based solely on in silico predictions.
The Kirby–Bauer disk diffusion test does not mention interpretative standards (e.g., CLSI VET08 or EUCAST).
Only eight antibiotics were tested; this panel is insufficient to fully characterize the resistance profile.
The correlation between genomic and phenotypic resistance (particularly for streptomycin) is plausible but not validated.
The Authors should provide details on the methodology (disc concentration, breakpoints) and, if possible, include MIC determination for the main antimicrobials.
The bglA-based PCR test is interesting, but the validation lacks rigour:
Specificity was tested on a very limited panel of species.
The claims regarding sensitivity (56 CFU/mL, 27.8 genome copies) are unusually high and should be verified with replicates and controls.
No field validation or comparison with existing multiplex PCR tests (Alexander et al., 2008; Guenther et al., 2008) is provided.
The Authors should include a more comprehensive validation dataset and discuss practical application (e.g., diagnostic utility, cost, and field feasibility).
Minor
Define abbreviations upon first use (e.g., CFU, HE, LD₅₀).
Include details on culture conditions (e.g., atmospheric CO₂ concentration).
Figures and tables are informative but could benefit from higher resolution and improved captions.
The species name Mannheimia glucosida should be written in italics throughout the text.
Ensure that gene names are consistent (e.g., lktA, plpD, adh in italics).
Comments on the Quality of English LanguageThe English is understandable but often literal and repetitive, suggesting a direct translation from Chinese.
The introduction is too long and unfocused; it could be shortened by removing descriptions reminiscent of those found in textbooks.
The discussion reiterates the results rather than interpreting them critically.
Substantial linguistic and structural revision is required, ideally by a native English-speaking microbiologist familiar with the Pasteurellaceae family.
Reviewer 2 Report
Comments and Suggestions for Authors
Dear authors,
I cannot help but express my utmost admiration for your work. This is the first time I have seen a manuscript so thoroughly developed, complete, and accurate at this stage of the review process. I am unable to offer any significant suggestions, as my comments would be purely stylistic (and therefore subjective) rather than truly substantive. For this reason, I will simply congratulate you on your excellent work.
Reviewer 3 Report
Comments and Suggestions for Authors
Dear authors,
Thank you for submitting this manuscript (mc) to this journal and we have read this with interest. But before I can advise the Editor to accept I have some questions to elucidate and some remarks to improve your mc.
major points: for me it is not clear are these sheep: lambs or adults, from a large flock housed or pastured, how big was the herd, presence of fever etc.
in the results presented, I see a lot of information that should be part of M&M, e.g. line 201-202, until isolation, line 227-228, line 235-238, the mouse model, so evalute this section very precisely for what is M&M and what is results.
minor points:
line 24: may be an ... in sheep in China
line 35 ... type, respiratory syncytial virus
line 42: note that M. haemolytica is seen now also as an important pathogen in adult cattle, see Biesheuvel et al., 2021 Vet. Journal
line 91-93: here is a lack of information, e.g. all adult and all from one flock, fever?? etc.
line 201: the prevalence is for me how frequent is it seen in a country or in a flock, so this paper is only about 1 flock in a part of China, may be somewhere high in the mountains or somewhere in the suburbs of a city, we do not know and this is not a paper about prevalence
The results part should be checked carefully and M&M and Result presented seperately
line 387-388 can be skipped, because it is repeated in the conclusion.
Round 2
Reviewer 1 Report
Comments and Suggestions for Authors
While the manuscript is much improved, the following minor points would further polish it:
Abstract and conclusion
The abstract now states M. glucosida "may be in important pathogen." This appears to be a typo. It should read "may be an important pathogen."
The conclusion is brief but adequate. It could be very slightly strengthened by reiterating the clinical relevance of the specific PCR and the novel genomic data.
Materials and Methods
For the animal model, consider adding a single sentence on the ethical justification for using mice instead of sheep (e.g., for preliminary pathogenicity screening under controlled conditions), which pre-empts a potential reader question.
Line 145 to 154: convert these data in a table.
Results
Figure 3 (Phylogenetic Trees): The text (lines ~275-279) states that the sodA gene is more variable and useful for differentiation. This is a key finding. Consider adding a sentence here, or in the discussion, on the practical implication—that sodA could be a better target for developing diagnostic assays compared to the more conserved 16S rRNA and infB genes.
Discussion
The sentence on lines 385-387: "It causes opportunistic infections..." is a bit abrupt. Consider linking it more smoothly, for example: "If confirmed as a pathogen, M. glucosida likely acts as an opportunistic agent, causing disease when host immunity is compromised..."
Technical corrections
Line 348: "IkA" should be "lktA" for consistency.
Line 367: "athe specific PCR" should be "a specific PCR".
Line 373: "5.6×107⁓5.6×100 CFU/ml" – Ensure the superscript formatting is correct (e.g., 10⁰).
Ensure all gene names are in italics consistently (e.g., lktA, sodA).
Reviewer 3 Report
Comments and Suggestions for Authors
Dear authors,
the paper has been improved seriously and I have only one serious remarkabout line 71:
kumar et al. ... a triple assay targeting specific genes of M. haemolytica , which ..
Nobody know what the other abbreviations mean, so laeve it!!
